# Characterisation of Drug-Induced Liver Injury in Patients with COVID-19 Detected by a Proactive Pharmacovigilance Program from Laboratory Signals

**DOI:** 10.3390/jcm10194432

**Published:** 2021-09-27

**Authors:** Ana Delgado, Stefan Stewart, Mikel Urroz, Amelia Rodríguez, Alberto M. Borobia, Ibtissam Akatbach-Bousaid, Miguel González-Muñoz, Elena Ramírez

**Affiliations:** 1Clinical Pharmacology Department, La Paz University Hospital-IdiPAZ, School of Medicine, Autonomous University of Madrid, 28046 Madrid, Spain; ana.delgadodemendoza@hotmail.com (A.D.); stefan.stewart@outlook.es (S.S.); mikel-el@hotmail.com (M.U.); amelia.rodriguez@salud.madrid.org (A.R.); alberto.borobia@salud.madrid.org (A.M.B.); 2Immunology Department, La Paz University Hospital-IdiPAZ, 28046 Madrid, Spain; iakatbachboussaid@gmail.com

**Keywords:** coronavirus disease 2019 (COVID-19), drug induced liver injury (DILI), pharmacovigilance, updated Roussel Uclaf Causality Assessment Method (RUCAM), lymphocyte transformation test (LTT)

## Abstract

Coronavirus disease 2019 (COVID-19) has a wide spectrum of clinical manifestations. An elevation of liver damage markers has been observed in numerous cases, which could be related to the empirical use of potentially hepatotoxic drugs. The aim of this study was to describe the clinical and analytical characteristics and perform a causality analysis from laboratory signals available of drug-induced liver injury (DILI) detected by a proactive pharmacovigilance program in patients hospitalised for COVID-19 at La Paz University Hospital in Madrid (Spain) from 1 March 2020 to 31 December 2020. The updated Roussel Uclaf Causality Assessment Method (RUCAM) was employed to assess DILI causality. A lymphocyte transformation test (LTT) was performed on 10 patients. Ultimately, 160 patients were included. The incidence of DILI (alanine aminotransferase >5, upper limit of normal) was 4.9%; of these, 60% had previous COVID-19 hepatitis, the stay was 8.1 days longer and 98.1% were being treated with more than 5 drugs. The most frequent mechanism was hepatocellular (57.5%), with mild severity (87.5%) and subsequent recovery (88.1%). The most commonly associated drugs were hydroxychloroquine, azithromycin, tocilizumab and ceftriaxone. The highest incidence rate of DILI per 10,000 defined daily doses (DDD) was with remdesivir (992.7/10,000 DDD). Some 80% of the LTTs performed were positive, with a RUCAM score of ≥4. The presence of DILI after COVID-19 was associated with longer hospital stays. An immune mechanism has been demonstrated in a small subset of DILI cases.

## 1. Introduction

In December 2019, a new coronavirus, Severe Acute Respiratory Syndrome Coronavirus 2 (SARS-CoV-2), emerged in Wuhan, Hubei Province, China. The epidemic disease caused by this coronavirus, known as coronavirus disease 2019 (COVID-19), has a wide spectrum of manifestations, ranging from asymptomatic infections to severe pneumonia, respiratory distress syndrome and death. Several studies have suggested that SARS-CoV-2 also affects the liver. Although various underlying mechanisms have been proposed, including hypoxia, COVID-19-associated hyperinflammatory syndrome and a direct cytopathic effect [1,2,3,4], drug-induced liver injury (DILI) should also be considered. Apart from remdesivir and dexamethasone, approved for COVID19 treatment, many drugs have been used empirically based on experience and availability. In addition, there are differences in baseline characteristics between patients. It is important that information is available on the possible adverse effects these drugs can cause so that they can be used as safely as possible [5]. A randomised controlled trial of lopinavir/ritonavir in patients with severe COVID-19 reported that elevated aspartate aminotransferase (AST) and alanine aminotransferase (ALT) levels occurred as an adverse effect in several patients [6]. A cross-sectional study of patients from Canton or Guandong, China, reported that 76.3% of patients experienced liver impairment, 21.5% from DILI. In this study, the use of lopinavir/ritonavir increased the risk of DILI (odds ratio 4.44 to 5.03, *p* < 0.01) [7]. Similarly, in a study of hospitalised patients with SARS-CoV-2 infection treated with remdesivir, 22% had an increase in liver enzymes [8]. The Spanish Pharmacovigilance System for Medicinal Products for Human Use, composed of the Autonomous Centres for Pharmacovigilance and coordinated by the Spanish Agency for Medicines and Health Products (AEMPS), is closely monitoring reports of adverse drug reactions (ADRs) in these patients. These drugs include hydroxychloroquine, lopinavir/ritonavir, azithromycin, ceftriaxone, amoxicillin-clavulanate and tocilizumab. Until 31 December 2020, 583 notifications of suspected ADRs had been registered by the AEMPS. Among them were 283 cases of liver disorders, in which hydroxychloroquine (126 cases) appears predominantly, followed by lopinavir/ritonavir (54), tocilizumab (49) and remdesivir (23) [9].

Therefore, the main objective of the study was to describe the clinical and analytical characteristics, severity, type, outcome, underlying factors and causality analysis of DILI in the COVID-19 population.

## 2. Materials and Methods

### 2.1. Setting

La Paz University Hospital in Madrid, Spain, is a tertiary-care teaching facility where, since 2007, all admissions to wards have been monitored by a Pharmacovigilance Program from Laboratory Signals in Hospital (PPLSH) [10]. We conducted a retrospective observational study using the medical records of patients treated for SARS-CoV-2 infection and hepatotoxicity signals from the PPLSH, during the period from 1 March 2020 to 31 December 2020. The study was approved by the Hospital’s Institutional Review Board (protocol PI-4505). Due to the retrospective nature of the study, the absence of informed consent was permitted. Inclusion criteria consisted of patients aged 18 years or older who, during their hospitalisation due to COVID-19 infection, had been treated with any drug and had a recorded hepatotoxicity signal. Patients who did not present a confirmed diagnosis of COVID-19 were excluded.

### 2.2. Hepatotoxicity Signal

Definition of automatic laboratory signal employed to detect DILI: ALT 5 times the upper limit of normal (ULN) [11].

### 2.3. Detection, Evaluation and Notification

The procedure for detecting and evaluating ADRs has been described elsewhere [10]. Briefly, in **phase I**, on-file laboratory data at admission or during hospitalisation were screened 7 days a week, 24 h a day, for ALT. In **phase II**, the patients were identified to avoid duplicates, and electronic medical records were reviewed. In those cases where ALT was clearly attributable to the primary diagnosis of COVID-19 or to other alternative causes, the patients were not further analysed because an ADR was unlikely. In **phase III**, a case-by-case evaluation was performed for the remaining cases (Figure 1). Regarding the evaluation of the drug cause versus the alternative non-drug-induced cause, we only considered a drug cause when there was no alternative cause to explain the signal and, for the patients with COVID-19, when liver function worsened while COVID-19 clinical and laboratory parameters improved. When a DILI was suspected, withdrawal of the suspected drugs was discussed with the attending physician, and the patient was followed-up during hospitalisation and referred to a pharmacovigilance consultation. For all patients categorised as having a DILI, a complete adverse reaction report was submitted to the pharmacovigilance centre in Madrid.

### 2.4. Causality Assessment

The causality assessment was performed using the Roussel Uclaf Causality Assessment Method (RUCAM) [11]. It contains specific conditions according to the type of hepatitis:Hepatocellular (ratio (R) ALT/alkaline phosphatase (AP) ≥ 5)Cholestatic (R ALT/AP ≤ 2) or Mixed (2 < R < 5).

For both types, the algorithm has 7 items: (1) time to onset from the beginning of the drug; (2) course of ALT or AP after cessation of the drug; (3) risk factors (alcohol use (gr/day) and age ≥55 years); (4) concomitant drug use; (5) search for alternative causes; (6) previous knowledge of drug hepatotoxicity; and (7) response to unintentional re-exposure. The total score classifies the event: ≤0, excluded; 1–2, unlikely; 3–5, possible; 6–8, probable; and ≥9, highly probable. We considered the categories of possible, probable or definite for drug-related reactions.

### 2.5. Collection of Patient Data

All information was retrospectively collected from the electronic medical record. We collected the patients’ demographic variables (sex, height, weight, country, date of birth) as well as comorbidities for both infection and hepatitis (e.g., history of ADR, previous liver disease, previous COVID hepatitis, high blood pressure, hypercholesterolaemia, diabetes mellitus I and II, toxic habits, pregnancy, chronic liver disease). Previous COVID hepatitis was considered if ALT was normalized before the start of treatment with the drug. The ADR history collected was the type of ADR and associated drugs. Hospitalisation data were also collected (date of admission, duration, outcome and date of outcome, admission to ICU, dates and outcome). Data were also collected on the symptoms and severity of the COVID-19 infection (pneumonia or other complications, including their severity, as well as the date of onset, the outcome and the maximum CURB-65 score for organ failure).

The laboratory variables were recorded at 3 time points (baseline, maximum or peak and outcome recovery) and included the following: albumin, ALT, AST, AP, total bilirubin, creatinine, gamma-glutamyl transferase (GGT), eosinophil count, glomerular filtration rate, lactate dehydrogenase, pH and prothrombin activity. Regarding unwanted effects, data were collected on hepatitis (previous liver function, function during admission and subsequent follow-up, symptoms, additional tests, whether treatment or medical care was required, duration and outcome). The type and severity of the DILI were categorised following the consensus of the International DILI Expert Group [11]. The drug causality of the hepatitis (updated RUCAM algorithm) scores were also collected. Polypharmacy was defined as the concomitant use of more than 5 drugs. Drug consumption was characterised by the defined daily dose (DDD), which is the assumed average maintenance dose per day for a drug used for its main indication in adults. DDD is assigned per Anatomical Therapeutic Chemical Classification System code and route of administration [12].

In some cases, a lymphocyte transformation test (LTT) was performed with the suspected drugs, as previously described [13]. The test was considered positive if the stimulation index defined as the ratio between the mean value of counts per minute in cultures with the drug and those obtained without the drug was ≥2.

### 2.6. Expected Sample Size and Basis for Its Determination

There was a total of 8719 hospitalised patients with COVID-19 infection in the study period. A sample of 152 individuals would be sufficient to estimate, with 95% confidence and a precision of ±5 percentage units, a population percentage with DILI that was expected to be approximately 10%. The percentage of necessary replacements was estimated to be 10%.

### 2.7. Data Analysis

The continuous variables were presented as means (standard deviation (SD)) and compared using an analysis of variance if the data were normally distributed, or as medians (interquartile range) and compared with the Kruskal–Wallis test if the data were not normally distributed. The chi-squared test was performed to compare sex distribution and morbidity variables. We used IBM SPSS Statistics version 20.0 (IBM Corporation, Armonk, NY, USA) for the statistical analysis.

## 3. Results

### 3.1. DILI Incidence

During the established period, 36,905 patients were hospitalised, of whom 8719 (23.6%) were diagnosed with COVID-19. In this same period, through automated laboratory signal detection by the Pharmacovigilance Programme of La Paz University Hospital (Phase I), 322,724 ALT results were obtained, of which 6653 (2.1%) met criteria for a laboratory signal.

The DILI incidence among the COVID-19 cases was 4.9%, compared with 3.7% in patients hospitalised for other causes, totalling 1471 patients. In this total, patients and alternative causes were identified (Phase II). A total of 351 (23.9%) cases of DILI were obtained, and a case-by-case evaluation was performed (Phase III). Ultimately, 160 (10.9%) patients with COVID-19 were found with DILI. The frequency of DILI in patients recovered from COVID-19 hepatitis was 36.2%. The flow chart is shown in Figure 1.

### 3.2. General Characteristics of the Cohort

Out of 160 patients ultimately detected, 77.5% (*n* = 124) were men. The mean age was 54.3 (SD 13.9) years. The general characteristics of the cohort are shown in Table 1. Regarding country of origin, the most frequent was Spain (89 patients; 55.6%), followed by Ecuador (19 patients; 11.9%). Mean weight (kg) was 82.7 (SD 17.2), mean body mass index (BMI) (kg/m^2^) was 28.1 (SD 5.5). Fifteen (9.4%) patients had previous liver disease, of which the majority (*n* = 14) had steatosis and only 1 had chronic hepatitis B. Ninety-six cases (60%) had had a previous COVID-19 hepatitis (*p* = 0.004). Regarding comorbidities, 49 (30.6%) patients had elevated blood pressure, 71 (44.4%) dyslipidaemia and 11 (6.8%) diabetes mellitus. Regarding toxic habits, 8.8% (*n* = 14) of the patients were smokers and 3.8% (*n* = 6) had an active alcohol habit. Of the patient cohort, 95.6% (*n* = 153) presented an episode of DILI during hospitalisation; in the remaining 7 cases, DILI caused a new admission. During the study period, the mean stay in the hospitalisation wards of non-COVID patients was 6.3 days. Patients who developed DILI during hospitalisation had a hospital stay 8.1 days longer than the mean hospital stay for patients with only COVID-19 (14.3 days). After hospitalisation, 101 (63.1%) patients were discharged, 39 (24.4%) were transferred to other intermediate care facilities, 1 (0.6%) developed sequelae and 19 (11.9%) died. Only 18 (11.3%) patients had a history of ADR, the most frequent type being allergy (17 cases) and the most frequent drug being non-steroidal anti-inflammatory drugs (5 cases) (Table 2).

### 3.3. Characteristics of DILI Cases

The most frequent type of DILI was hepatocellular, presented by 92 (57.5%) patients; nevertheless, 26.2% (*n* = 42) of the cases could not be classified because they did not present an AP value in the clinical history. A total of 11.3% of hepatitis cases became chronic. The severity of the cases was mostly defined as mild (87.5%; *n* = 140) and was fatal in only 1 (0.6%) case. Recovery was the most common outcome after the episode, achieved by 141 (88.1%) patients. Levels of ALT increased 13.3 (SD 22.0) times above the ULN, while AP was elevated 1.3 (SD 1.6) times and total bilirubin 1.0 ± 2.0 times. Hypertransaminasaemia was recorded in the clinical records in 125 (78.1%) patients, whereas in 41 (25.6%) cases the ADR was recorded in the discharge medical records.

During the stay, the mean number of administered drugs per patient was 14.7 (SD 7.6); 157 (98.1%) were treated with polypharmacy (>5 drugs). In the drug causality analysis (RUCAM), 51.2% (*n* = 82) of cases were scored as probable (at least 1 drug assessed with a score ≥6 points) and 48.8% (*n* = 78) as possible (at least 1 drug assessed with a score ≥3 points). Table 3 shows the characteristics of these patients.

### 3.4. Culprit Drugs

Out of 1308 drugs used, 263 were considered related to DILI. The most commonly related drugs were hydroxychloroquine (82 cases), azithromycin (56), ceftriaxone (35) and tocilizumab (33), followed by remdesivir (14), paracetamol (11), enoxaparin (9) and ritonavir/lopinavir (7). Table 4 shows the characteristics of DILI cases by related drugs (RUCAM score ≥ 3). The higher ALT elevation observed with enoxaparin was statistically significant. Similarly, the higher AP elevation using remdesivir was statistically significant.

The highest incidence rate of DILI per 10,000 DDDs was due to treatment with remdesivir (992.7/10,000 DDDs), followed by azithromycin (211.1/10,000 DDDs), hydroxychloroquine (195.1/10,000 DDDs) and ritonavir/lopinavir (137.8/10,000 DDDs) (Table 5).

### 3.5. Lymphocyte Transformation Test Results and Concordance with RUCAM

An LTT was performed on 10 patients, of whom 8 (80%) obtained positive results for a suspected drug; hydroxychloroquine (3) and azithromycin (3) were the most frequent.

All LTT-positive drugs had a RUCAM score ≥4. On the other hand, 28 negative results were obtained, of which 7 had a RUCAM score ≥6 (Table 6).

## 4. Discussion

### 4.1. Incidence and Length of Stay

The DILI definition of the Pharmacovigilance Program based on laboratory signals was made in accordance with the criteria defined by the DILI Expert Working Group [11]. In our study, the incidence of DILI cases in patients with COVID-19 was higher than in patients admitted for other causes (4.9% vs. 3.7%). A study performed in China with 217 patients diagnosed with COVID-19 reported 94 ADRs, of which 30 (13.8%) were liver disorders. In the same study, the length of hospital stay in the group with ADRs compared with the non-ADR group was significantly longer (21.1 days vs. 15.9 days, *p* < 0.001) [14]. These results are in agreement with our study, which showed a prolongation of hospitalisation of 8.1 days.

Various studies have revealed that patients infected with SARS-CoV, Middle East Respiratory Syndrome-CoV, and SARS-CoV-2 can develop varying degrees of liver injury. Patients with SARS had primarily manifested with mild and moderate elevation of ALT and/or AST during the early stage of the disease. Some patients had shown a reduction in serum albumin and an increase in serum bilirubin levels, with severe cases more likely to have severe liver injury compared with mild cases [15,16,17]. In our study, 265 (18%) COVID-19 inpatients had ALT > 5 ULN. In a retrospective study, 37.2% of patients had elevated liver enzymes on admission, defined as liver function parameters (ALT, AST, AP, GGT, total bilirubin) above the ULN; the alteration was associated with more fever and higher C-reactive protein and procalcitonin alteration as well as previous use of ritonavir/lopinavir [18]. Differences in the definition of liver function impairment (>1 ULN vs. >5 ULN) could account for the differences in frequency compared with our study.

### 4.2. Characteristics of the Cohort

In our study, 60% of the patients with DILI had previously had COVID-19-induced liver injury. The underlying mechanisms of liver injury in patients with COVID-19 can include psychological stress, systemic inflammatory cytokine response and progression of preexisting liver diseases (i.e., simple fatty liver disease to steatohepatitis) [19]. The mean BMI in our study was 28.1 kg/m^2^. Patients with obesity often have underdiagnosed fatty liver disease associated with metabolic dysfunction and are at increased risk of severe COVID-19 disease. In addition, they are more likely to be hospitalised and receive the necessary treatments to treat acute respiratory distress syndrome and systemic inflammation, which together with the presence of fatty liver and sometimes non-alcoholic steatohepatitis, can predispose patients to DILI [20]. In our study, the mean age of the patients was 54.3 (SD 13.9) years. In a study performed by Cai et al., (2020) [21], the median age in patients who had altered liver enzymes or liver damage (defined as ALT or AST > 3 ULN or GGT or total bilirubin > 2 ULN) was 53 (42–64) years, with a value of *p* = 0.04, compared with those who did not develop liver alterations, whose median age was 47 (33–59). In another study with the same pharmacovigilance program, older men experienced greater DILI severity, which was associated with higher alcohol and low albumin levels in this group [22]. The frequency of other metabolic risk factors (diabetes mellitus, hypertension, dyslipidaemia) in the DILI cohort was no higher than in the hospital’s COVID-19 cohort [23]. These results are in agreement with other studies [4,24].

Many of the drugs used to treat COVID-19 are potentially hepatotoxic, and their use in the form of polypharmacy added to the proinflammatory state particularly increases the risk of DILI [20]. In our study, 98.1% of patients had been treated with five or more drugs. Other studies among patients with COVID-19 by the Hospital Pharmacovigilance System found that the number of drugs administered to patients who developed ADRs was statistically significantly higher than in patients who did not develop an adverse reaction (8.57 ± 3.34 vs. 5.40 ± 2.10; *p* < 0.001) [14]. Concomitant use of multiple drugs is often shown to control comorbid conditions and improve efficacy. This concomitant use of multiple drugs (more than 5 concomitant drugs) has been defined as “polypharmacy”. Polypharmacy has been associated with adverse consequences such as higher health care costs and increased risk of adverse drug-drug interaction reactions, among other things [25]. Along these lines, a retrospective study on the possible hepatotoxicity of ritonavir/lopinavir in patients with COVID-19 noted that ritonavir, being a strong C3A4 inhibitor, could promote hepatic toxicity from azithromycin [26].

### 4.3. Culprit Drugs

Remdesivir had the highest incidence rate of DILI per administration in our study. The hepatotoxicity of remdesivir has been subject to debate. Although randomized trials in COVID-19 demonstrate equivalent liver enzyme elevations between treatment and control groups [27], screening of WHO safety reports database showed that, of the total 387 reports of ADRs for this drug, 130 (34%) were hepatic, whereas the majority (79; 61%) were hepatobiliary [28]. Azithromycin had the second highest incidence rate of DILI in this study. Large clinical trials using azithromycin reported acute, transient, asymptomatic increases in serum ALT levels in 1–2% of patients [29]. Azithromycin has been associated with DILI in patients with COVID-19 infection, with <6 ULN ALT elevations in 40% of patients [20]. In a prospective study by the US DILI consortium, azithromycin was associated with DILI in patients with pre-existing liver disease compared with those without liver disease (6.7% vs. 1.5%, *p* = 0.006) [30].

Tocilizumab, interleukine-6 receptor antagonist, has been proposed to treat severe forms of COVID-19 because interleukine-6 plays an important role in COVID-19 induced cytokine storm. Elevation of liver enzymes in rheumatoid arthritis patients treated with tocilizumab is a well-documented phenomenon [31]. A first case of DILI associated with the use of tocilizumab in a COVID-19 patient described an increase of serum transaminase levels of 40-ULN in a healthy 52-years-old man. Nevertheless, tocilizumab had a positive effect on clinical and laboratory parameters in cytokine storm, with transaminases values normalizing in 10 days [32]. A study in which seven patients were followed found that six had liver damage values within a normal range or slightly altered; after receiving tocilizumab during admission, all had ALT elevation five times the ULN [33]. Regarding treatment with lopinavir/ritonavir, the study performed by Fan et al. (2020) [18] showed that more patients with impaired liver function had received this treatment (57.8%) compared with those with normal liver function (31.3%; *p* = 0.01). In another study, 65 hospitalised patients with severe COVID-19 and treatment with this drug were studied; 25 (38.5%) of them developed liver impairment [26]. Of the protease inhibitors regimens studied, the greatest risk of DILI has been observed among patients receiving full-dose ritonavir. Similarly, hepatitis B and/or C virus coinfection has been associated with a greater risk of DILI compared with those with HIV-infected patient with no hepatitis [34].

A review of the FDA adverse drug reports for the period January to July 2020 for azithromycin, chloroquine and hydroxychloroquine showed a rise in reports from 592 before to 2492 after their emergency use authorization for COVID-19 infection, mostly for hydroxychloroquine (596 cases) and azithromycin (184 cases), the second most frequent event being hepatitis [35]. In Brazil, between March and August 2020, a total of 631 adverse event reports in 402 patients with COVID-19 were received, of which 56 cases (9%) were hepatic, including 28 attributed to hydroxychloroquine and 4 to azithromycin [36]. Similarly, a study on patients with COVID-19 with altered liver enzymes showed how treatment with hydroxychloroquine, remdesivir, lopinavir/ritonavir and tocilizumab were associated with an elevation of ALT five times the ULN [4]. It has been found that the presence of a mild background inflammation could enhance the liver induced by drugs (e.g., azithromycin, chloroquine) that are not hepatotoxic at the same doses when administered to control animals [37,38,39]. One of the proposed mechanisms for enhancing drug-induced hepatotoxicity during inflammation could be associated with the production of reactive drug metabolites by inflammatory cells. Mieloperoxidase is an enzyme present in inflammatory cells, such as macrophages and neutrophils. Myeloperoxidase is able to convert some drugs to reactive cytotoxic metabolites [40,41].

### 4.4. Lymphocyte Transformation Test in the Causal Diagnosis of DILI

DILI is a diagnosis of exclusion that is under-recognised and under-reported. Even when the diagnosis is made, it is often difficult to determine which of the possible drugs is the cause of the liver damage [42]. Causality algorithms can help in diagnosing the causative drug. A RUCAM-based assessment has shown high sensitivity (86%), specificity (89%), positive predictive value (93%) and negative predictive value (78%) [43]. The RUCAM had been used to assess DILI-causing drugs in hospitalised patients at a single medical centre in Korea. Antidepressants, antihistamines and antibacterials were the most common hepatotoxicity-causing drugs [44]. However, RUCAM has poor discrimination when used in polypharmacy settings, as in our case (98.1% polypharmacy). LTT has been proposed as a diagnostic method to determine whether a patient has been sensitised to a specific drug; it has been widely used in Japan for the diagnosis of DILI [45,46].

Eighty per cent of the LTTs performed were positive for a suspected drug, the most frequent being hydroxychloroquine and azithromycin. All the drugs that obtained a positive outcome had a RUCAM score ≥4. Considering the limitation of the small number of patients tested, LTT could be an in vitro test that would support the diagnosis of immune-mediated DILI. Furthermore, LTT could guide treatment with alternative drugs in cases of DILI.

### 4.5. Limitations

This study’s main limitation was that the evaluation of causality of a possible DILI does not completely rule out the influence of COVID-19. However, a DILI was only considered when there was a dissociation between clinical improvement and worsening of ALS. In addition, a minimum follow-up of 6 months has made it possible to evaluate the recovery or sequelae of these DILIs and to study a possible DILI immune mechanism.

## 5. Conclusions

The incidence of DILI in patients with COVID-19 was higher than in patients hospitalised for other causes. The presence of DILI after COVID-19 was associated with longer hospital stay. Most cases were mild, hepatocellular in mechanism and with subsequent recovery. We observed that a high frequency of previous COVID hepatitis was observed in DILI.

The most frequently associated drugs were hydroxychloroquine, azithromycin, tocilizumab, ceftriaxone, lopinavir/ritonavir, paracetamol, remdesivir and enoxaparin, with RUCAM causality analysis defined as probable in 51.2% of cases. The highest incidence rate of DILI per 10,000 DDDs was for remdesivir (992.7/10,000 DDDs). An immune mechanism has been demonstrated in DILI in a small subset of DILI cases.

## Figures and Tables

**Figure 1 jcm-10-04432-f001:**
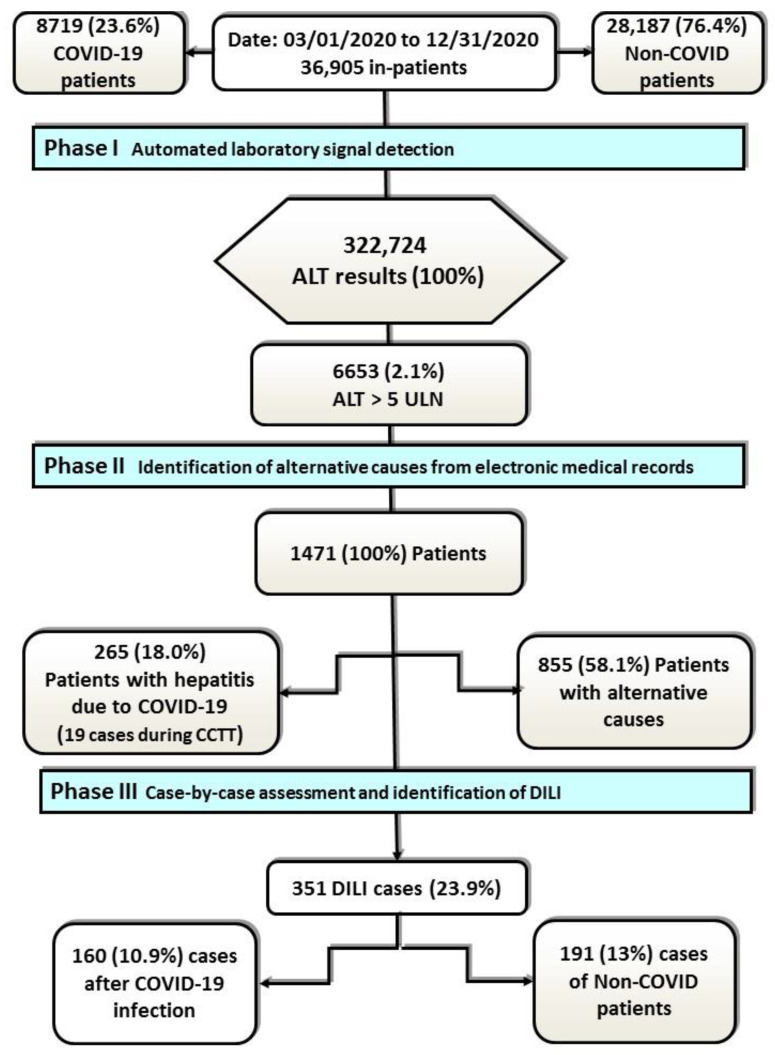
Flowchart. Abbreviations: ALT, alanine aminotransferase; DILI, drug-induced liver injury; CCTT, clinical trials; ULN, upper limit of normal.

**Table 1 jcm-10-04432-t001:** Characteristics of cohort.

**Variable**			
**Number of cases, *n***		160	
**Age (years), mean (SD)**		54.3	(13.9)
**Sex (male), *n* (%)**		124	(77.5)
**Country of origin, *n* (%)**	Spain	89	55.6
	Ecuador	19	11.9
	Peru	8	5.0
	Philippines	6	3.7
	Others	38	23.8
**Number of drugs, mean (SD)**		14.7	(7.6)
**Polypharmacy *, *n* (%)**		157	(98.1)
**History of ADR, *n* (%)**	No	142	(88.7)
	Yes	18	(11.3)
**Previous liver disease, *n* (%)**	No	145	(91.6)
	Yes Steatosis Hepatitis B chronic	15 14 1	(9.4)
**Previous COVID hepatitis (ALT > 5 ULN)**	No	64	(40.0)
	Yes	96	(60.0)
**Weight (kg), mean (SD)**		82.7	(17.2)
**Height (cm), mean (SD)**		168.2	(9.01)
**Serum albumin (g/dL, NR: 2.9–5.2), mean (SD)**		3.6	(0.64)
**BMI (kg/m^2^), mean (SD)**		28.1	(5.5)
**Hypertension, *n* (%)**	No	111	(69.4)
	Yes	49	(30.6)
**Dyslipidaemia, *n* (%)**	No	89	(55.6)
	Yes	71	(44.4)
**Diabetes mellitus, *n* (%)**	No	149	(93.2)
	Yes	11	(6.8)
**Smoking habit, *n* (%)**	No	301	(81.2)
	Smoker	14	(8.8)
	Former	16	(10.0)
**Alcoholic habit, *n* (%)**	No	153	(95.6)
	Alcoholism	6	(3.8)
	Former	1	(0.6)
**Drug abuse habit, *n* (%)**	No	159	(99.4)
	Yes	1	(0.6)
**CURB-65, *n* (%)**	0	51	(31.9)
	1	49	(30.6)
	2	42	(26.3)
	3	1	(0.6)
	4	0	(0.0)
	5	0	(0.0)
	Unknown	17	(10.6)
**ICU stay**	Total (%)	38	(23.8)
	Discharge *n* (% ICU)	26	(68.4)
	Death *n* (% ICU)	12	(31.6)
**Outcome of hospitalisation**	Discharge	101	(63.1)
	Transfer ^#^	39	(24.4)
	Death	19	(11.9)
	Sequelae	1	(0.6)

Polypharmacy *, >5 concomitant drugs; ^#^ Transfer to an intermediate care facility. Abbreviations: ADR, adverse drug reaction; BMI, body mass index; CURB-65, Confusion, Urea nitrogen, Respiratory rate, Blood pressure, 65 years old; DILI, drug-induced liver injury; NR, normal range; SD, standard deviation.

**Table 2 jcm-10-04432-t002:** History of ADRs.

DRUG	
NSAIDS	ALLERGY
NSAIDS, METAMIZOLE, PENICILLIN, ACETYLSALICYLIC ACID	ALLERGY
AZITHROMYCIN, MACROLIDES, NSAIDS	ALLERGY
CIPROFLOXACIN	ALLERGY
CHLOROQUINE	ALLERGY
DILTIAZEM	ALLERGY
HALOPERIDOL	ALLERGY
IBUPROFEN	ALLERGY
QUETIAPINE	INTOLERANCE
ISONIAZID	ALLERGY
METAMIZOLE	ALLERGY
METAMIZOLE	ALLERGY
METRONIDAZOLE	ALLERGY
PENICILLIN, TETRACYCLINE, CONTRAST AGENT	ALLERGY
TETRACYCLINE	ALLERGY
TRAMADOL	ALLERGY
TRAMADOL	ALLERGY
VANCOMYCIN, BETA-LACTAMS	ALLERGY

**Table 3 jcm-10-04432-t003:** Characteristics of DILI cases.

**Variable**					
**Number of cases, *n***		160			
**Type, *n* (%)**	Hepatocellular	92	(57.5)		
	Mixed	20	(12.5)		
	Cholestatic	6	(3.8)		
	Not classified	42	(26.2)		
**RUCAM classification, *n* (%)**	Highly probable	0	(0.0)		
	Probable	82	(51.2)		
	Possible	78	(48.8)		
**Severity, *n* (%)**	Mild	140	(87.5)		
	Moderate	11	(6.9)		
	Severe	8	(5.0)		
	Fatal	1	(0.6)		
**Outcome, *n* (%)**	Recovery	141	(88.1)		
	Transplant	0	(0.0)		
	Death	1	(0.6)		
	No associated death	18	(11.2)		
**Chronification of hepatitis, *n* (%)**	No chronification	117	(83.0)		
	Chronification	16	(11.3)		
	Unknown	8	(5.7)		
**Recorded HT in DR, *n* (%)**	No	35	(21.9)		
	Yes	125	(78.1)		
**Recorded DILI in DR, *n* (%)**	No	119	(74.4)		
	Yes	41	(25.6)		
**Laboratory Parameters**		**Value**		**Number of Times ULN**	
		**Mean**	**SD**	**Mean**	**SD**
**ALT**, U/L (NR < 35)	Baseline	47.3	22.3	1.1	0.6
	Maximum	465.8	769.0	13.3	22.0
	Recovered	197.4	766.0	5.6	21.8
**LDH**, U/L (NR, 100–190)	Baseline	374.4	149.1	1.9	0.8
	Maximum	886.7	2059.9	4.6	10.8
	Recovered	585.9	2114.4	3.1	11.1
**AP**, U/L (NR, 46–116)	Baseline	97.9	56.8	0.8	0.5
	Maximum	150.7	184.6	1.3	1.6
	Recovered	102.6	97.0	0.7	0.8
**Creatinine**, mg/dL (NR, 0.7–1.30)	Baseline	0.9	0.3	0.7	0.2
	Maximum	1.1	0.9	0.8	0.7
	Recovered	0.9	0.5	0.7	0.4
**Total bilirubin**, mg/dL (NR, 0.3–1.2)	Baseline	0.7	0.3	0.6	0.3
	Maximum	1.2	2.4	1.0	2.0
	Recovered	1.0	2.3	0.8	1.9
**GGT**, U/L (NR < 73)	Baseline	99.4	128.5	1.4	1.8
	Maximum	357.3	360.7	4.9	4.9
	Recovered	95.5	122.2	1.3	1.7
**TPAC**, (%) (NR, 70–120)	Baseline	94.8	17.3	1.4	0.2
	Maximum	98.4	26.1	1.4	0.4
	Recovered	94.2	19.9	1.3	0.3
**pH** (7.35–7.45) ^#^	Baseline	7.42	0.07	1.0	0.01
	Maximum	7.33	0.19	1.0	0.03
	Recovered	7.33	0.14	1.0	0.02
**Eosinophils**, 10³/μL (NR, 0.02–0.65) ^#^	Baseline	0.10	0.13	0.15	0.19
	Maximum	0.02	0.08	0.03	0.08
	Recovered	0.11	0.13	0.12	0.12

^#^ Number of times ULN; Abbreviations: ADR, adverse drug reaction; ALT, alanine aminotransferase; AP, alkaline phosphatase; DR, discharge records; DILI, drug-induced liver injury; GGT, gamma glutamyl transferase; LDH, lactate dehydrogenase; SD, standard deviation; TPAC, thromboplastin activity; ULN, upper limit of normal.

**Table 4 jcm-10-04432-t004:** Characteristics of DILI cases for the most common drugs (RUCAM score ≥ 3).

Variable		Azithromycin		Hydroxychloroquine/Chloroquine		Ceftriaxone		Tocilizumab		Remdesivir		R/Lopinavir		Paracetamol		Enoxaparin		*p-*Value
**Number of patients, *n***		56		82		35		33		14		7		11		9		
**Age (years), mean (SD)**		56.6	12.2	57.3	11.5	54.1	11.8	57.7	7.6	50.7	13.3	53.3	10.1	46.0	12.7	58.1	10.8	<0.001
**Sex (male), *n* (%)**		41	73.2	64	78.0	29	82.9	24	68.6	12	85.7	6	85.7	7	63.6	9	100	0.199
**Hospital stay, mean (SD)**		19.8	22.2	18.8	22.1	14.9	10.5	22.5	18.8	19.8	17.1	16.4	9.9	16.1	4.5	75.4	129.7	<0.001
**History of ADR, *n* (%)**		5	8.9	10	12.2	2	5.7	3	9.1	14	100	1	14.3	1	9.1	2	22.2	0.421
**Type, *n* (%)**	Hepatocellular	29	51.8	47	57.3	19	54.2	19	57.6	6	42.9	4	57.1	7	63.6	6	66.7	0.531
	Mixed	5	8.9	6	7.3	3	8.6	1	3.0	1	7.1	1	14.3	1	9.1	2	22.2	
	Cholestatic	3	5.4	3	3.7	3	8.6	2	6.1	1	7.1	1	14.3	1	9.1	0		
	Not Classified	19	33.9	26	31.7	10	28.6	11	33.3	6	42.9	1	14.3	2	18.2	1	11.1	
**Number of drugs, mean (SD)**		13.1	4.6	13.3	4.8	11.9	3.8	15.7	6.0	13.4	8.4	14.1	3.5	14.0	2.8	12.9	5.8	0.650
**Polypharmacy *, *n* (%)**		55	98.2	81	98.8	34	97.1	33	100	14	100	7	100	10	90.9	9	100	0.433
**RUCAM classification, *n* (%)**	Probable	17	30.4	32	39.0	13	37.1	11	33.3	6	42.9	2	28.6	1	9.1	4	44.4	0.373
	Possible	39	69.6	50	61.0	22	68.9	22	66.7	8	57.1	5	71.4	10	90.9	5	55.6	
**Severity, *n* (%)**	Mild	53	94.6	75	91.5	32	91.4	31	94.0	12	85.7	7	100	11	100	8	88.9	0.416
	Moderate	3	5.4	5	6.10	2	5.7	1	3.0	0	0.0	0	0.0	0	0.0	0	0.0	
	Severe	0	0.0	2	2.4	1	2.9	1	3.0	2	14.3	0	0.0	0	0.0	1	11.1	
**Outcome, *n* (%)**	Recovered	51	91.0	73	89.0	33	94.2	28	84.9	14	100	7	100	11	100	8	88.9	0.006
	Death	0	0.0	0	0.0	0	0.0	0	0.0	0	0.0	0	0.0	0	0.0	0	0.0	
	No associated death	4	7.0	8	9.8	1	2.9	5	15.15	0	0.0	0	0.0	0	0.0	1	11.1	
	Sequelae	1	2	1	1.2	1	2.9	0	0.0	0	0.0	0	0.0	0	0.0	0	0.0	
**Previous liver disease, *n* (%)**		7	12.5	9	11	3	8.6	4	12.1	1	7.2	2	28.6	1	9.1	2	22.2	0.524
**Weight (kg), mean (SD)**		79.2	18.3	78.6	19.1	83.4	15.1	78.6	8.3	82.8	12.6	68.0	4.2	78.0		93.3	9.9	<0.001
**Height (cm), mean (SD)**		167.8	10.2	167.1	9.0	170.8	10.0	170.3	7.8	170.4	7.5	159.5	0.7	178.0		170.3	7.4	<0.001
**Serum albumin (g/dL), mean (SD)**		3.7	0.7	3.6	0.7	3.7	0.8	3.4	0.6	3.6	0.4	4.0	0.4	3.8	0.4	3.6	0.9	0.611
**BMI (kg/m^2^), mean (SD)**		25,8	5.8	26.3	5.8	27.2	3.9	26.9	3.4	28.9	4.4	26.7	1.9	24.6		32.4	5.4	<0.001
**Hypertension, *n* (%)**		16	28.6	26	21.7	11	31.4	6	18.2	4	28.6	3	42.9	2	18.2	2	22.2	0.357
**Dyslipidaemia** **, *n* (%)**		28	50	40	48.8	16	45.7	20	60.6	3	21.4	5	71.4	6	54.5	4	44.4	0.332
**DM, *n* (%)**		2	3.6	5	6.1	2	5.7	0	0.0	0	0.0	2	28.6	0	0.0	1	11.1	0.481
**Smoking Habit, *n* (%)**	No	45	80.4	64	78.0	32	91.4	30	90.9	11	78.6	6	85.7	8	72.7	6	66.7	0.341
	Smoker	4	7.1	8	9.8	1	2.9	1	3.0	2	14.3	1	14.3	2	18.2	1	11.1	
	Former	7	12.5	10	12.2	2	5.7	2	6.1	1	7.1	0	0.0	1	9.1	2	22.2	
**Alcoholic habit, *n* (%)**	No Alcoholism Former	53	94.4	79	96.3	34	97.1	32	97.0	13	92.9	7	100	10	91.0	9	100	0.608
		2	3.4	2	2.5	1	2.9	1	3.0	1	7.1	0	0.0	1	9.0	0	0.0	
		1	1.8	1	1.2	0	0.0	0	0.0	0	0.0	0	0.0	0	0.0	0	0.0	
**Hepatitis chronification, *n* (%)**		1	1.8	2	2.4	0	0.0	0	0.0	0	0.0	0	0.0	0	0.0	2	22.2	0.427
**Recorded HT in DR, *n* (%)**		42	75	62	75.6	26	74.3	27	81.8	12	85.7	6	85.7	10	91.0	8	88.9	0.755
**Recorded ADR in DR, *n* (%)**		15	26.8	26	31.7	11	31.4	11	33.3	4	28.6	4	57.1	4	36.4	3	33.3	0.893
**ALT**, U/L (NR < 35)	Baseline	39.6	23.4	41.5	23.8	36.1	20.2	36.0	17.9	36.2	21.5	45.0	15.3	45.4	26.6	40.2	27.8	0.001
	Maximum	331.7	297.9	383.5	427.1	301.8	111.0	559.2	1240.0	339.9	207.9	292.9	188.3	298.4	103.7	813.4	1514.3	<0.001
	Recovered	89.2	262.7	141.1	434.9	70.7	67.9	315.6	1286.7	91.8	110.6	72.8	76.4	60.6	57.7	573.2	1599.8	0.002
**LDH**, U/L (NR, 100–190)	Baseline	384.0	152.7	364.7	154.5	377.1	129.1	413.9	159.4	372.6	129.4	248.3	97.4	356.3	120.4	266.0	94.1	<0.001
	Maximum	613.1	498.6	786.6	1409.6	519.6	208.1	755.9	657.1	563.3	343.9	452.9	193.6	590.6	253.2	2655.3	6938.4	<0.001
	Recovered	298.3	491.5	524.1	1496.2	237.4	86.0	393.1	703.1	202.7	35.6	197.3	28.3	194.1	36.2	2533.8	6983.4	<0.001
**AP**, U/L (NR, 46–116)	Baseline	94.0	62.0	86.7	60.7	110.5	76.4	74.9	14.0	157.0	105.0	71.7	20.9	76.0	14.9	88.1	21.3	<0.001
	Maximum	123.0	90.1	128.4	109.7	148.6	112.6	94.4	44.8	314.3	509.6	109.0	60.0	105.4	67.2	117.4	60.8	<0.001
	Recovered	79.2	22.9	78.5	23.8	92.7	39.0	77.7	20.7	131.7	46.1	80.7	30.5	69.4	12.8	89.0	25.3	<0.001
**Cr**, mg/dL (NR, 0.7–1.30)	Baseline	0.9	0.4	0.9	0.4	0.9	0.2	0.8	0.2	0.9	0.2	0.9	0.2	0.8	0.2	0.9	0.2	<0.001
	Maximum	0.9	0.7	1.1	0.8	0.9	0.5	1.1	0.9	0.7	0.4	1.1	0.4	0.8	0.3	1.3	1.2	<0.001
	Recovered	0.9	0.5	1.0	0.5	0.9	0.3	1.0	0.7	0.7	0.3	0.9	0.2	0.8	0.2	1.1	0.9	<0.001
**TB**, mg/dL (NR, 0.3–1.2)	Baseline	0.6	0.3	0.6	0.2	0.6	0.2	0.7	0.4	0.6	0.2	0.7	0.2	0.7	0.3	0.5	0.1	<0.001
	Maximum	1.0	0.7	1.1	0.9	0.8	0.5	1.2	1.0	0.7	0.8	1.0	0.6	1.1	0.7	1.0	0.3	<0.001
	Recovered	0.7	0.5	0.8	0.7	0.6	0.3	0.9	0.8	0.6	0.2	0.6	0.2	0.7	0.2	0.9	0.3	<0.001
**GGT**, U/L (NR < 73)	Baseline	99.1	126.2	95.1	119.9	100.1	97.7	84.4	92.1	97.4	135.7	53.3	18.0	98.0	105.2	73.2	76.1	<0.001
	Maximum	373.0	402.6	367.0	384.5	409.7	368.3	357.0	410.6	343.2	295.7	425.3	457.7	544.0	518.1	219.4	136.9	<0.001
	Recovered	81.6	113.6	99.5	139.2	108.6	121.2	89.6	136.3	95.2	66.4	101.8	121.7	78.4	90.8	63.3	25.1	<0.001
**TPAC**, (%) (NR, 70–20)	Baseline	90	19	90	19	91	16	92	18	108	9	87	29	94	11	90	11	<0.001
	Maximum	97	23	99	23	100	21	89	31	113	7	97	25	99	24	95	30	<0.001
	Recovered	96	16	93	18	97	12	91	23	95	15	110	7	91	13	84	28	0.001
**pH** (7.35–7.45)	Baseline	7.4	0.1	7.4	0.1	7.4	0.0	7.4	0.1	7.5	0.0			7.4	0.1	7.5	0.1	<0.001
	Maximum	7.3	0.2	7.3	0.1	7.4	0.1	7.3	0.2	7.4	0.1			7.4	0.2	7.3	0.2	<0.001
	Recovered	7.4	0.1	7.3	0.1	7.4	0.0	7.3	0.2	7.4	0.0			7.4	0.1	7.3	0.2	<0.001
**Eo**, 10³/μL NR, 0.02–0.65)	Baseline	0.1	0.1	0.1	0.1	0.1	0.2	0.1	0.1	0.0	0.0	0.2	0.2	0.1	0.1	0.2	0.1	0.0321
	Maximum	0.0	0.0	0.0	0.1	0.0	0.1	0.0	0.0	0.0	0.0	0.1	0.3	0.0	0.0	0.0	0.0	0.312
	Recovered	0.1	0.1	0.2	0.2	0.2	0.2	0.1	0.1	0.1	0.1	0.3	0.3	0.1	0.1	0.1	0.1	0.067

Polypharmacy *, >5 concomitant drugs; *p*-value was the level of significance of the chi-squared test for discrete variables or of analysis of variance or the Kruskal–Wallis test, as appropriate, for continuous variables. Abbreviations: ADR, adverse drug reaction; ALT, alanine aminotransferase; AP, alkaline phosphatase; BMI, body mass index; Cr, creatinine; DILI, drug-induced liver injury; DM, diabetes mellitus; DR, discharge records; Eo, eosinophils; GGT, gamma-glutamyl transferase; HT, hypertransaminasaemia; LDH, lactate dehydrogenase; NR, normal range; SD, standard deviation; TB, total bilirubin; TPAC, thromboplastin activity (%).

**Table 5 jcm-10-04432-t005:** Incidence by consumption in cases of in-hospital DILI.

DRUG	Cases	ATC Code	DDD Value (U) Route	Consumption in DDDs in DILI DH *	Consumption During the Study Period ^#^ (DDDs)	Incidence Rate ^&^ (Per 10,000 DDDs)	95% CI (Per 10,000 DDDs)
**Remdesivir**	14	J05AB16	0.1 (g) P	109.2	1100	992.7	932.2–1055.7
**Azithromycin**	56	J01FA10	0.5 (g) P	194.4	9207	211.1	184.4–241.5
**Hydroxychloroquine**	82	P01BA02	0.516 (g) O	336.5	17,245	195.1	169.5–224.4
**Ritonavir/lopinavir**	7	J05AR10	0.8 (g) O	24.6	1785	137.8	115.9–162.0
**Tocilizumab**	33	L04AC07	20 (mg) P	76.5	9920	77.1	61.7–96.2
**Ceftriaxone**	35	J01DD04	2 (g) P	148.5	23,586	63	48.4–80.6
**Enoxaparin**	9	B01AB05	2 (TU) P	170.2	107,660	15.8	9.1–24.7
**Paracetamol**	11	N02BE01	3 (g) O/P/R	176	219,410	8.0	3.5–15.8

* Consumption in DDD in DILI DH = (total grams used/DDD value in grams) × (number of cases). ^#^ Consumption During the study periods = total gram used during the study period/DDD value in grams. ^&^ Incidence rate = (Consumption in DDD in DILI DH)/(Consumption during the study period) × 10,000. Abbreviations: ATC, Anatomical, Therapeutic, Chemical classification system; CI, confidence interval; DDD, defined daily dose; DH, during hospitalisation; O, oral; P, parenteral; R, rectal; (U), Unit.

**Table 6 jcm-10-04432-t006:** RUCAM concordance with LTT.

	Drug 1		Drug 2		Drug 3		Drug 4		Drug 5		Drug 6	
Code	LTT	RUCAM Score	LTT	RUCAM Score	LTT	RUCAM Score	LTT	RUCAM Score	LTT	RUCAM Score	LTT	RUCAM Score
**02**	Hydroxychloroquine		Lopinavir/Ritonavir		Ceftriaxone							
	(−)	+3	(+)	+4	(−)	+1						
**03**	Lopinavir/ Ritonavir		Interferon beta-1b		Levofloxacin		Dexketoprofen		Hydroxychloroquine			
	(−)	+2	(−)	+4	(+)	+4	(−)	+2	(−)	+4		
**04**	Tocilizumab		Hydroxychloroquine									
	(+)	+6	(−)	+6								
**06**	Azithromycin											
	(+)	+6										
**08**	Azithromycin		Hydroxychloroquine		Lopinavir/ Ritonavir		Ceftriaxone		Pantoprazole			
	(−)	+5	(+)	+6	(−)	+5	(−)	+6	(+)	+4		
**09**	Azithromycin		Hydroxychloroquine		Tocilizumab		Paracetamol		Metamizole			
	(−)	+6	(−)	+6	(−)	+6	(−)	+3	(−)	+5		
**10**	Azithromycin		Hydroxychloroquine		Tocilizumab		Paracetamol					
	(+)	+4	(+)	+4	(−)	+4	(−)	+3				
**13**	Levofloxacin		Azithromycin		Hydroxychloroquine		Tocilizumab					
	(−)	+6	(−)	+4	(−)	+4	(−)	+7				
**17**	Hydroxychloroquine		Ceftriaxone		Piperacillin/Tazobactam		Metamizole		Paracetamol		Lopinavir/Ritonavir	
	(+)	(+4)	(+)	+4	(+)	+4	(−)	+3	(−)	4	(−)	+4
**106**	Hydroxychloroquine		Azithromycin		Doxycycline		Dexketoprofen		Enoxaparin		Omeprazole	
	(−)	+4	(+)	+4	(−)	+4	(+)	+4	(−)	+4	(−)	+4

LTT, lymphocyte transformation test; (−) Negative; (+) Positive. RUCAM scores: ≤0, excluded; 1–2, unlikely; 3–5, possible; 6–8, probable; ≥9, highly probable.

## Data Availability

The data that support the findings of this study are available from the corresponding author upon reasonable request.

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
