# Peer review of "Characterisation of Drug-Induced Liver Injury in Patients with COVID-19 Detected by a Proactive Pharmacovigilance Program from Laboratory Signals"

_jcm, 2021, doi:10.3390/jcm10194432_

Round 1

Reviewer 1 Report

As the number of cases treated with sCOVID-19 increases, so does the number of cases of drug-induced liver injury. To the best of this reviewer's knowledge, there is no comprehensive report on drug-induced liver injury caused by COVID-19 treatment, so we consider this report to be of great value.

Minor

  1. Literature numbers should be written in Arabic numerals (digit), not Roman numerals.
  2. Table 4-6 is complicated and difficult to understand, so it should be presented in a clear way.
  3. There have already been many reports of drug-induced liver injury caused by various therapeutic agents against COVID-19. The authors should cite as many of the most recent papers as possible.

Author Response

Comments and Suggestions for Authors

As the number of cases treated with sCOVID-19 increases, so does the number of cases of drug-induced liver injury. To the best of this reviewer's knowledge, there is no comprehensive report on drug-induced liver injury caused by COVID-19 treatment, so we consider this report to be of great value.

Authors: We would like to thank to the reviewer 1 for their comments and suggestions for the manuscript. We have carefully revised the manuscript accordingly to the reviewer´s suggestions. Below you can see the respond to the reviewer´s comments. The changes in the original manuscript have been made using the track changes mode in MS Word, and we enclosed a “clean” version with the changes accepted.

Minor

  1. Literature numbers should be written in Arabic numerals (digit), not Roman numerals.

Authors: Done

  1. Table 4-6 is complicated and difficult to understand, so it should be presented in a clear way.

Authors: We try to clarify tables 4-6.

  1. There have already been many reports of drug-induced liver injury caused by various therapeutic agents against COVID-19. The authors should cite as many of the most recent papers as possible.

Authors: We update the reports of DILI in COVID-19 patients. We added thirteen more references to the discussion, culprit drugs section.

Regards,

Elena Ramírez

Miguel González-Muñoz

Reviewer 2 Report

This paper looks at the DILI in COVID-19 patients. The authors evaluated the ADRs identified by a computer program and applied the RUCAM causality algorithm. The paper brings new information on the DILI of various drugs used in the context of COVID-19 infection. While hard to limit the causality of COVID-19 and drugs, the authors found a method to separate the 2 which is not entirely reliable, but plausible.

The strengths of the paper are the method of patient selection, the use of the RUCAM algorithm and also the LTT analysis. Also, the calculation of the incidence reported to the DDDs is an interesting addition. As a drawback, the paper contains a lot of data (please see comments on the Table 3 and 4), which unfortunately is not always organized for an increased clarity. Please find below my comments and suggestions.

  1. In some parts of the paper (e.g. flowchart) patients are referred as “after COVID-19 infection” and in others “during their hospitalization due to COVID-19 infection”. I suppose the later is the correct one, please adjust
  2. Please have a look at the DDD definition, it is more like the most often used daily dose and is set for research purposes only and not a standard for treatment
  3. In the RESULTS section/DILI incidence you mention “Of the patient cohort, 95.6% (n=153) presented an episode of DILI during hospitalisation.”. I was under the impression that all patients were hospitalized. Were the rest of 5% presenting with DILI at admission? please Also, for the following sentence: “The mean hospital stay was 6.3 days.” Please clarify for which patients was this hospital stay. Is a bit confusing here, as in the next sentence and in the table are different numbers (22 days)
  4. Table 1: the variable “Lengthened hospital stay, days (SD)” could be taken out, it is presented in the text and here in the table is not clear to what is lengthen the hospital stay
  5. Table 1: for the variable “Previous COVID-19 hepatitis (ALT > 5 ULN)” please clarify if the previous covid hepatitis was during the same covid infection or previous covid infection. Depending on this, this could be a risk factor if the ALT were normalized before the drug start. If not, definitely is an alternative cause. Maybe the authors could include more information on how they treated the patients in this group in the Methods section. Did you calculate maybe if the difference of 10% is significant? This would be interesting to see
  6. Table 3: The “Number of times ULN” section is not really of interest for all the variables, but only for the liver function. So maybe include the information as an additional column above (enumerating the parameters twice in the same table does not increase the clarity) or keep in this table only the parameters that we are usually calculating the number of times ULN.
  7. Table 4: this table contains a lot of information, maybe not all is needed (like 3 points in time pH for every drug, or Drug abuse which is 0 for all drugs, or the RUCAM classification, n (%)-Highly probable row etc). I would suggest maybe to separate the patient characteristics from reaction's characteristics in this table. I would keep the characteristics of DILI here, with the results of RUCAM only and maybe separately include a table of the statistically significant characteristics. Also, please include an explanation on what comparison was the p-value calculated.
  8. Table 5: please include explanation on how was calculated Consumption in DDDs in DILI DH*, Consumption During the study period (DDDs) and Incidence rate (per 10,000)
  9. Discussion: 1st paragraph---(15.9±7.52 days vs. 21.1±8.63)---- I think these numbers are to be presented the other way around
  10. Conclusion: maybe I missed that, but I could not see that “The presence of DILI during hospitalisation for COVID-19 was associated with greater severity, …… and polypharmacy.” Indeed, for the hospital stay is true. Another thing that I could not find in the Results, was “The highest incidence and severity were related to being male and having a high BMI,…”. To me, these risk factors should have been analyzed in a regression analysis to be able to say this.

Author Response

Comments and Suggestions for Authors

This paper looks at the DILI in COVID-19 patients. The authors evaluated the ADRs identified by a computer program and applied the RUCAM causality algorithm. The paper brings new information on the DILI of various drugs used in the context of COVID-19 infection. While hard to limit the causality of COVID-19 and drugs, the authors found a method to separate the 2 which is not entirely reliable, but plausible.

The strengths of the paper are the method of patient selection, the use of the RUCAM algorithm and also the LTT analysis. Also, the calculation of the incidence reported to the DDDs is an interesting addition. As a drawback, the paper contains a lot of data (please see comments on the Table 3 and 4), which unfortunately is not always organized for an increased clarity.

Authors: We would like to thank to the reviewer 2 for their comments and suggestions for the manuscript. We have carefully revised the manuscript accordingly to the reviewer´s suggestions. Below you can see the respond to the reviewer´s comments. The changes in the original manuscript have been made using the track changes mode in MS Word, and we enclosed a “clean” version with the changes accepted.

Please find below my comments and suggestions.

1. In some parts of the paper (e.g. flowchart) patients are referred as “after COVID-19 infection” and in others “during their hospitalization due to COVID-19 infection”. I suppose the later is the correct one, please adjust

 Authors: Corrected. Thank you.

2. Please have a look at the DDD definition, it is more like the most often used daily dose and is set for research purposes only and not a standard for treatment.

 Authors: Drug consumption was characterised by the defined daily dose (DDD), which is the assumed average maintenance dose per day for a drug used for its main indication in adults. DDD is assigned per Anatomical Therapeutic Chemical Classification System code and route of administration. We try to clarify this point in the manuscript.

 3. In the RESULTS section/DILI incidence you mention “Of the patient cohort, 95.6% (n=153) presented an episode of DILI during hospitalisation.”. I was under the impression that all patients were hospitalized. Were the rest of 5% presenting with DILI at admission? please Also, for the following sentence: “The mean hospital stay was 6.3 days.” Please clarify for which patients was this hospital stay. Is a bit confusing here, as in the next sentence and in the table are different numbers (22 days)

Authors: Of the patient cohort, 95.6% (n = 153) presented an episode of DILI during hospitalization; in the remaining 7 cases, DILI caused a new admission. During the study period, the mean stay of non-COVID patients was 6.3 days. Patients who developed DILI during hospitalisation had a hospital stay 8.1 days longer than the mean hospital stay for patients with only COVID-19 (14.3 days). We try to clarify these points in the manuscript, Result section. These results presented in the text have been eliminated from table 1.

 4. Table 1: the variable “Lengthened hospital stay, days (SD)” could be taken out, it is presented in the text and here in the table is not clear to what is lengthen the hospital stay

 Authors: Done.

 5. Table 1: for the variable “Previous COVID-19 hepatitis (ALT > 5 ULN)” please clarify if the previous covid hepatitis was during the same covid infection or previous covid infection. Depending on this, this could be a risk factor if the ALT were normalized before the drug start. If not, definitely is an alternative cause. Maybe the authors could include more information on how they treated the patients in this group in the Methods section. Did you calculate maybe if the difference of 10% is significant? This would be interesting to see

 Authors: Thank you very much; we appreciate very much these comments. The variablePrevious COVID hepatitis”· was considered if ALT was normalized before drug start (n = 96 DILI cases). We try to clarify this point in the Method section, Collection of patient data paragraph. In phase II, of the procedure for detecting and evaluating DILI, the patients were identified to avoid duplicates, and electronic medical records were reviewed. In those cases where ALT was clearly attributable to the primary diagnosis of COVID-19 (265 (18%) patients with hepatitis due to COVID-19, Figure 1) or to other alternative causes (855 (58.1%) patients, Figure 1), the patients were not further analysed because an ADR was unlikely. We also try to clarify this point in Method section. Ninety-six cases (60%) had had a previous covid-19 hepatitis (p = 0.004). We add this analysis to the text of the results. We also add a discussion of this point.

6. Table 3: The “Number of times ULN” section is not really of interest for all the variables, but only for the liver function. So maybe include the information as an additional column above (enumerating the parameters twice in the same table does not increase the clarity) or keep in this table only the parameters that we are usually calculating the number of times ULN.

 Authors: We add an additional column above for “Number of time ULN” for Lab. parameters. Changes are not marked in the table 3.  

 7. Table 4: this table contains a lot of information, maybe not all is needed (like 3 points in time pH for every drug, or Drug abuse which is 0 for all drugs, or the RUCAM classification, n (%)-Highly probable row etc). I would suggest maybe to separate the patient characteristics from reaction's characteristics in this table. I would keep the characteristics of DILI here, with the results of RUCAM only and maybe separately include a table of the statistically significant characteristics. Also, please include an explanation on what comparison was the p-value calculated.

 Authors: We eliminated the rows “Drug abuse”, “RUCAM Highly probable” and "No" of comorbilities (changes are not marked). We propose to eliminate the row “DILI during hospitalization” (track changes marked). We also add an explanation for p-value calculation in the footnote of the table “p-value was the level of significance of the chi-squared test for  discrete variables or of analysis of variance or the Kruskal-Wallis test, as appropriate, for continuous variables”.

 8. Table 5: please include explanation on how was calculated Consumption in DDDs in DILI DH*, Consumption During the study period (DDDs) and Incidence rate (per 10,000)

 Authors: Done. We add to the footnote:

  • Consumption in DDD in DILI DH = (total grams used/DDD value in grams)*(number of cases).
  • Consumption During the study periods = total gram used during the study period /DDD value in grams.
  • Incidence rate = (Consumption in DDD in DILI DH)/(Consumption during the study period)*10,000.

9. Discussion: 1st paragraph---(15.9±7.52 days vs. 21.1±8.63)---- I think these numbers are to be presented the other way around

 Authors: Done.

 10. Conclusion: maybe I missed that, but I could not see that “The presence of DILI during hospitalisation for COVID-19 was associated with greater severity, …… and polypharmacy.” Indeed, for the hospital stay is true. Another thing that I could not find in the Results, was “The highest incidence and severity were related to being male and having a high BMI,…”. To me, these risk factors should have been analyzed in a regression analysis to be able to say this.

 Authors: We are very sorry for the errors. We corrected the conclusion in this sense. Thank you.

Regards,

Elena Ramírez

Miguel González-Muñoz